Workshop at the 6th Symposium on Advances in Approximate Bayesian Inference (non-archival), 2024 1–10

# On Uncertainty Quantification for Near-Bayes Optimal Algorithms

**Ziyu Wang**                                                    WZY196@GMAIL.COM
**Chris Holmes**                                          CHOLMES@STATS.OX.AC.UK
*Department of Statistics, University of Oxford*

## 1. Introduction

Bayesian modelling represents an important approach that enables favourable predictive performance in the small-sample regime and allows for the quantification of predictive uncertainty which is vital for high-stakes applications. Yet for many machine learning (ML) algorithms it can be difficult to derive or implement their Bayesian counterpart. For example, the development of Bayesian neural network (NN) methods encounters challenges with inference (Sun et al., 2018; Wang et al., 2018) and model misspecification (Aitchison, 2020; Fortuin et al., 2021); AutoML algorithms (Karmaker et al., 2021) involve complex processes for hyperparameter tuning and model aggregation which are hard to replicate in a Bayesian framework; and when algorithms are provided as black-box services (e.g., OpenAI, 2023), adapting them to Bayesian principles becomes impossible.

How can we bring back the benefits of the Bayesian paradigm without being limited by its traditional constraints? In this work we present a promising approach towards this challenge based on the following **basic postulation**: the ML algorithm of interest has competitive average-case performance on hypothetical datasets—or *tasks*—from a possibly *unknown task distribution* $\pi$, and our present task can be viewed as a random sample from the same $\pi$. Formally, suppose the algorithm $\mathcal{A}$ maps a training dataset $z_{1:n}$ to a *parameter estimate* $\mathcal{A}(z_{1:n})$; we require it to satisfy an inequality of the following form,

$$\mathbb{E}_{\theta_0 \sim \pi} \mathbb{E}_{(z_{1:n}, z_*) \sim \mathbb{P}_{\theta_0}} \ell(\mathcal{A}(z_{1:n}), z_*) \leq \inf_{\mathcal{A}'} \mathbb{E}_{\theta_0 \sim \pi} \mathbb{E}_{(z_{1:n}, z_*) \sim p_{\theta_0}} \ell(\mathcal{A}'(z_{1:n}), z_*) + \epsilon_n. \tag{1}$$

In the above, $\theta_0$ is a *parameter* that determines the data generating process $p_{\theta_0}$ in an ML task; $(z_{1:n}, z_*)$ denote the training and test samples; $\ell(\theta, z)$ denotes the loss function; and $\mathcal{A}'$ ranges over all possible algorithms that maps $z_{1:n}$ to an $\mathcal{A}'(z_{1:n}) \approx \theta_0$.

To understand this postulation, imagine a practitioner working on a new image classification dataset. To understand the suitability of a certain algorithm $\mathcal{A}$ (which may be a combination of an NN model, optimisation and hyperparameter tuning recipes), it is natural for them to start by reviewing past literature that evaluated $\mathcal{A}$ on datasets deemed similar to the present one. At a high level, the past and present tasks can be viewed as i.i.d. samples from the distribution $\pi$, and promising reports from past literature will provide evidence that (1) holds with a smaller $\epsilon_n$. The practitioner may then commit to the choice of $\mathcal{A}$ that

---

. This is an extended abstract formatted in accordance with the workshop guidelines. The full paper is available at this url.

corresponds to the smallest $\epsilon_n$. As another type of example, condition (1) is also relevant in *multi-task learning* scenarios, where it often appears as an assumption of the algorithm (e.g., Pentina and Lampert, 2014; Riou et al., 2023). *Foundation models* (Bommasani et al., 2021) that are pretrained on a diverse mix of datasets can also be viewed as optimised for (1), with a choice of $\pi$ designed to align with the downstream task of interest.

Algorithms that satisfy (1) are *near-Bayes optimal*, and the knowledge of the Bayesian posterior defined by $\pi$ would enable the minimisation of (1) (Ferguson, 1967). As exemplified above, there are many scenarios where it is more reasonable to assume knowledge of a near-optimal $\mathcal{A}$ than that of a *correctly specified* $\pi$. Yet with such a choice of $\mathcal{A}$, there remain the challenge of uncertainty quantification, and the risk of overfitting in a small-sample regime: for example, for regular parametric models maximum likelihood estimation (MLE) can be asymptotically near-Bayes optimal (Van der Vaart, 2000), but it does not provide any (epistemic) uncertainty estimate, and the predictive performance of MLE in the small-sample regime may well be improvable.

To address these issues, we build on the ideas of Fong et al. (2024); Holmes and Walker (2023) and investigate *martingale posteriors* (MPs; See §2 for a review). As our first contribution, we prove that when the algorithm $\mathcal{A}$ defines an approximate martingale, satisfies (1) and additional conditions, we can use $\mathcal{A}$ to construct an approximate MP which will provide a good approximation for the Bayesian posterior defined by $\pi$ in a Wasserstein sense (§3). Such results allow us to draw from the benefits of the latter without requiring explicit knowledge of $\pi$. Our results cover high-dimensional models and the pre-asymptotic regime; in this aspect it demonstrates advantages over classical approaches such as bootstrap aggregation (Breiman, 1996).

As a further contribution, in §4 we present MP-inspired algorithms for uncertainty quantification. Our method is based on sequential applications of a base estimation algorithm and applies to both NN and non-NN algorithms. We illustrate the method using a wide range of base algorithms, including Gaussian process models, boosting trees, AutoML algorithms, and diffusion models. In all experiments the proposed method outperforms competitive baselines such as bootstrap and deep ensemble (Lakshminarayanan et al., 2017).

## 2. Background

We briefly review the idea of martingale posteriors (Fong et al., 2024) which serve as the basis of our work, and refer readers to the full paper for a full introduction.

**From Bayesian to martingale posteriors.** Suppose we have i.i.d. observations $z_{1:n} \in \mathcal{Z}^{\otimes n}$ and access to a suitable algorithm $\bar{\mathcal{A}}$ which, for any $j \geq n$, maps any $j$ observations $z_{1:j} \in \mathcal{Z}^{\otimes j}$ to a possibly random parameter $\bar{\mathcal{A}}(z_{1:j}) \in \Theta$. For simplicity, suppose for the moment that any parameter $\theta \in \Theta$ indexes a distribution $p_\theta$ over $\mathcal{Z}$ which we can sample from.[1] Consider a sequence of data and parameter samples defined as follows:

$$\widehat{\theta}_j := \bar{\mathcal{A}}(z_{1:n} \cup \widehat{z}_{n+1:j}), \quad \widehat{z}_{j+1} \sim p_{\widehat{\theta}_j}, \quad \text{for } j = n, n+1, \dots \tag{2}$$

---

1. As we explain in the full paper, this can be extended to supervised learning where $z = (x, y)$ and $\theta$ only indexes a conditional distribution $p_\theta(y \mid x)$, and our theory and methodology will continue to apply.

Intuitively, for any reasonable $\bar{\mathcal{A}}$, we expect the resulted $\{\widehat{\theta}_j : j > n\}$ to converge almost surely w.r.t. a *suitably chosen (semi-)metric $d$*, because after observing infinite samples the parameter uncertainty should vanish.[2] Denote this stochastic limit as $\widehat{\theta}_\infty$, then the variation in the distribution $p(\widehat{\theta}_\infty \mid z_{1:n})$ reflects the uncertainty due to the missing observations $\{z_j\}_{j=n+1}^\infty$ and thus measures *epistemic uncertainty* (Der Kiureghian and Ditlevsen, 2009).

As we review in the full paper, the above formulation is justified in part through the fact that it generalises Bayesian posteriors: the latter can be recovered by defining $\bar{\mathcal{A}}(z_{1:j})$ to sample from a Bayesian posterior $\pi(\theta \mid z_{1:j})$, assuming Doob's theorem (Doob, 1949) applies (e.g., if the posterior mean is bounded w.r.t. a certain vector semi-norm $\|\cdot\|$). More generally, as long as an algorithm is such that $\{\mathbb{E}_{\widehat{\theta}_j \sim \bar{\mathcal{A}}(z_{1:j})}\widehat{\theta}_j\}_{j=n}^\infty$ form a martingale and is bounded w.r.t. $\|\cdot\|$, by Doob's theorem the limit $\widehat{\theta}_\infty$ exists. The distribution $p(\widehat{\theta}_\infty \mid z_{1:n})$ is hence called a *martingale posterior* (MP).

**Martingales for machine learning?** The MP framework requires an explicitly and correctly specified prior, instead requiring our prior knowledge be expressible in the form of an algorithm $\bar{\mathcal{A}}$, but it also requires $\bar{\mathcal{A}}$ to define a martingale. Recent works have explored purposely constructed $\mathcal{A}$ (Fong et al., 2024; Lee et al., 2022; Ghalebikesabi et al., 2023) to satisfy this requirement, but it is unclear how common ML algorithms can be adapted for this goal. In this work we bridge this gap, building on the observation that online gradient descent (GD) defines a valid MP (Holmes and Walker, 2023): for

$$\widehat{\theta}_{j+1} := \widehat{\theta}_j + \eta_j \nabla_\theta \log p_{\widehat{\theta}_j}(\widehat{z}_{j+1}), \quad \text{where } \widehat{z}_{j+1} \sim p_{\widehat{\theta}_j}, \tag{3}$$

we have $\mathbb{E}(\widehat{\theta}_{j+1} \mid z_{1:j}) = \widehat{\theta}_j$. We start from the observation that a natural gradient variant of (3) connects to sequential MLE and can be near Bayes-optimal. This algorithm, along with its regularised variant, provide the motivating examples for our theory and methodology.

Another unaddressed question is how general MPs can be justified theoretically, beyond the somewhat vague belief that the imputations from a suitable $\bar{\mathcal{A}}$ may "approximate the missing data well". While previous works established consistency for specific MPs (Fong et al., 2024; Holmes and Walker, 2023), such results do not provide justification for the uncertainty estimates. Moreover, the intuition that imputations approximate the missing data well is somewhat flawed in the small-sample regime, where uncertainty quantification is most needed. Our theoretical analysis will address this question.

## 3. Analysis of Martingale Posteriors with Near-Optimal Algorithms

**Informal summary of the main result.** Our analysis applies in simplified scenarios to algorithms that are near-Bayes optimal in the sense of (1) and satisfy additional conditions. A stronger version of its assumptions can be summarized as follows:

(A1) We have an *online*, deterministic estimation algorithm which defines

$$\widehat{\theta}_{j+1} := \widehat{\text{Alg}}_{j+1}(\widehat{\theta}_j, \widehat{z}_{j+1}), \quad \widehat{z}_{j+1} \sim p_{\widehat{\theta}_j}. \tag{2'}$$

$\{\widehat{\theta}_j\}$ approximately forms a martingale w.r.t. some vector semi-norm $\|\cdot\|$.

---

2. For overparameterised models the semi-metric $d(\theta, \theta')$ should only measure the difference between $p_\theta$ and $p_{\theta'}$; this is also related to previous works on "function-space inference" (e.g., Sun et al., 2018).

(A2) $\|\theta - \theta'\|$ dominates the 2-Wasserstein distance $W_{2,\|\cdot\|_z}(p_\theta, p_{\theta'})$ defined by some $\|\cdot\|_z$.

(A3) For all $l > n$, let $\breve{\theta}_l$ be the estimate obtained by applying $\{\widehat{\mathrm{Alg}}_j\}$ to $l$ samples from the prior predictive distribution: $z_{1:l} \sim p_{\theta_0}, \theta_0 \sim \pi$. Introduce $\bar{\varepsilon}^2_{B,l} := \mathbb{E}_{z_{1:l}, \theta_0 \sim \pi} \|\mathbb{E}(\theta_0 \mid z_{1:l}) - \theta_0\|^2$, and $\breve{\varepsilon}^2_{ex,l} := \mathbb{E}\|\breve{\theta}_l - \theta_0\|^2 - \bar{\varepsilon}^2_{B,l}$. Then $\breve{\varepsilon}^2_{ex,l} \lesssim l^{-s}\bar{\varepsilon}^2_{B,l}$ for some $s > 0$.

(A4) (i) The function $\widehat{\Delta}_j(\theta, z) := \widehat{\mathrm{Alg}}_j(\theta, z) - \theta$ is $O(\eta_j)$-Lipschitz continuous w.r.t. both $(\theta, \|\cdot\|)$ and $(z, \|\cdot\|_z)$ where $\eta_j \asymp j^{-1}$. (ii) $\mathbb{E}_{z \sim p_{\theta'}} \widehat{\Delta}_j(\theta, z') = \eta_j H_{\theta,j}(\theta' - \theta) + O(\|\theta' - \theta\|^2)$.

In the above, (A3) formalises the condition (1), and (A4) are stability conditions; for GD algorithms, $\eta_j$ may correspond to the step-size. We also introduce miscellaneous conditions which generally hold for parametric models and the asymptotic ($n \gg \dim \theta$) regime, but also in the nonasymptotic examples below. We refer readers to the full paper for a formal statement and discussion of all assumptions and the result.

Under the above conditions we prove the following 2-Wasserstein distance bound between the unknown Bayesian posterior $\pi_n$, and the MP $\hat{p}_{mp,n}$:

$$\boxed{\mathbb{E}_\pi W^2_{2,\theta}(\pi_n, \hat{p}_{mp,n}) \ll \bar{\varepsilon}^2_{B,n}.} \tag{4}$$

As $\bar{\varepsilon}^2_{B,n}$ above also equals the expected squared radius of $\pi_n$ under $\|\cdot\|$, the result essentially states that *the difference between the MP and the Bayesian posterior a higher-order term.* It then follows that the MP can provide meaningful uncertainty estimates on new datasets sampled from $\pi$, which is especially interesting when explicit knowledge of $\pi$ is not available.

**Examples.** While the theorem does not cover deep NN models, it justifies similar algorithms on examples that cover high-dimensional models and the small-sample regime. We summarize the examples below and refer readers to the full paper for derivations and details.

(E1) Let $p_\theta$ be an exponential family. Then a sequential MLE algorithm has the form of (A1), and will satisfy (A1), (A3) and (A4) if $n \gtrsim \sqrt{\dim \theta}$ and $\pi$ is a conjugate prior which is not "too concentrated". (A2) needs to be verified on a case-by-case basis; we show it holds for e.g. Gaussian, exponential and Bernoulli families. Note that the result applies when the true parameter is not yet identified ($n \ll \dim \theta$).

(E2) Let $\pi$ be the iso-Gaussian process over a Hilbert space $\mathcal{H}$,[3] and $A : \mathcal{H} \to \mathcal{Z}$ be a Hilbert-Schmidt operator. Suppose $p_\theta(z) = \mathcal{N}(z \mid A\theta, I)$ where $\mathcal{N}_{\mathcal{Z}}$ denotes the shifted iso-Gaussian process on $\mathcal{Z}$. Define $\|\theta - \theta'\| = \|(A^\top A)^{1/2}(\theta - \theta')\|_{\mathcal{H}}$, and let $\widehat{\mathrm{Alg}}_j$ be defined through sequential maximum-a-posteriori (MAP) estimation. Then our theorem will always apply, providing a bound of $\mathbb{E}W^2_{2,\theta}(\pi_n, \hat{p}_{mp,n}) = \mathcal{O}(\bar{\varepsilon}^2_{B,n}/n)$. Note how this result does not depend on the extrinsic dimensionality of $\theta$.

As we explain in the full paper, (E2) above is closely related to Gaussian process (GP) regression which is similarly an infinite-dimensional linear inverse problem.[4] The same

---

3. See Skorohod (2012). $\pi$ is the white noise process if $\mathcal{H}$ is an $L_2$ space, or standard Gaussian if $\mathcal{H} = \mathbb{R}^d$.
4. For example, note $\pi = \mathcal{GP}(0, k)$ (the GP prior) if $\mathcal{H}$ is the reproducing kernel Hilbert space defined by $k$.

principle of sequential MAP estimation leads to the following algorithm, which can be easily implemented if we use random feature approximations for $\mathcal{H}$:

$$\widehat{\theta}_{j+1} := \arg\min_{\theta \in \mathcal{H}} \ \sum_{i=1}^{j}(f_{\widehat{\theta}_j}(x_i) - f_\theta(x_i))^2 + (f_\theta(x_{j+1}) - y_{j+1})^2 + n^{-1}\|\theta - \widehat{\theta}_j\|_{\mathcal{H}}^2. \quad (5)$$

While conjugate GP inference is a well-studied problem, previous works on Bayesian neural networks (BNNs) often justify their method by analysing similar random feature models (e.g., Osband et al., 2018; He et al., 2020) due to their connection to wide NN models (Lee et al., 2019). From our perspective, (5) is also interesting because it connects to a proximal Bregman objective (Bae et al., 2022) we will use for DNN models in the following.

## 4. MP-Inspired Uncertainty Quantification for General ML Algorithms

In §3 we have demonstrated the efficacy of the MP (2') instantiated with sequential MLE or its regularised variant. It may be reasonable to expect that similar methods are broadly applicable beyond the scope of our analysis. We thus consider a general method as follows:

---

1. Initialisation: $D_n := z_{1:n}, \widehat{\theta}_n \leftarrow \mathcal{A}_0(D_n)$

2. for $j \leftarrow n, n+1, \ldots, n + \lfloor N/\Delta n \rfloor$

   (a) Sample $\widehat{z}_{n_j:n_j+\Delta n} \sim p_{\widehat{\theta}_j}; D_{j+1} \leftarrow D_j \cup \widehat{z}_{n_j:n_j+\Delta n}$

   (b) $\widehat{\theta}_{j+1} \leftarrow \mathcal{A}(D_{j+1}; \widehat{\theta}_j)$

3. Repeat 1–2 for $K$ times; use the resulted $\{\widehat{\theta}_{n+\lfloor N/\Delta n \rfloor}^{(k)}\}_{k=1}^{K}$ to form an ensemble

---

In the above, $(\mathcal{A}_0(D), \mathcal{A}(D; \theta))$ denote a general point estimation algorithm; our analysis loosely suggests that any algorithm may be used if it is efficient in the sense of (1). We allow $\mathcal{A}(D; \theta)$ to resume from the previous-iteration optima $\theta$ if possible, but it may also refit a predictor from scratch. Compared with (2'), we also modify the algorithm to process $\Delta n > 1$ samples at each iteration. The resulted ensemble enables us to quantify epistemic uncertainty and improve predictive performance of the base algorithm.

**Remark 1 (comparison with bootstrap)** *The above algorithm has a similar form to the parametric bootstrap (Efron, 2012), which defines an ensemble by repeatedly drawing $n$ samples from an initial $p_{\widehat{\theta}_n}$ and computing a parameter estimate sole on these samples. In comparison, our method retains the original dataset $\{z_{1:n}\}$ and may be justified as approximating MPs if we consider $\Delta n \to 1, N \to \infty$. In the full paper we provide two specific examples where our algorithm has a more desirable theoretical behaviour than parametric and nonparametric bootstrap. We will also compare to these approaches empirically.*

**Remark 2 (comparison with Fong et al. 2024)** *The resampling scheme above resembles that in Fong et al. (2024). A key distinction is that we allow for base algorithms that do not satisfy the coherence condition in Fong et al. (2024, §3.2); the coherence condition required the algorithm $\mathcal{A}$ to possess a notion of epistemic uncertainty by itself and implied $\mathcal{A}$ can already be viewed as "implicitly Bayesian" (Mlodozeniec et al., 2024; see also Berti et al., 2021). In contrast, we allow the use of point estimation algorithms such as MLE,*

as justified by the analysis in §3, and uses the resampling scheme to augment *them with a coherent notion of epistemic uncertainty. It is nonetheless interesting to note that the same method can be applied to both estimation and Bayesian prediction algorithms and, in contrast to standard bootstrap approaches, will not inflate the predictive uncertainty in the latter case* (Fong et al., 2024).

**Remark 3 (a modified objective for DNNs)** *Many ML algorithms can be plugged into our method directly, but for DNN-based algorithms it can be beneficial to modify them to model the effect of early stopping: while DNNs are often trained to minimise a (regularised) empirical risk, due to early stopping $\widehat{\theta}_j$ will typically not reach the optima of its objective w.r.t. $D_j$; when processing new samples, it can be desirable to avoid further optimisation on the old $D_j$. We thus follow Bae et al. (2022) and modify the empirical risk by replacing the loss for $D_j$ with a function-space Bregman divergence; see our full paper for details. As a special case, for square loss, the new objective has the form of* (5) *(minus regularisation).*

## 5. Experiments

We evaluate the proposed method empirically across a variety of tasks. For space reasons, we summarise the findings below and defer setup details and full results to the full paper.

**Synthetic multi-task learning experiments.** We first complement the theoretical results in §3 with numerical simulations, on a setup inspired by previous theoretical works on multi-task learning (Tripuraneni et al., 2020; Du et al., 2020; Wang et al., 2022): we have $m$ i.i.d. pretraining tasks, each with $n_{pret}$ samples, and a test task with $n_*$ observations sampled from the same $\pi$. In each task we generate $y \mid x, \theta \sim \mathcal{N}(0, f_\theta(\Phi_0(x)))$ where $f_\theta \sim \mathcal{GP}(0, \bar{k})$ with $\bar{k}$ being an RBF kernel and $\Phi_0$ defined by a DNN. Both $\bar{k}$ and $\Phi_0$ are unknown to the user, who constructs an MP using (5), with $\mathcal{H}$ defined by applying the kernel learning algorithm in Wang et al. (2022) to the pretraining dataset. Therefore, all assumptions in §3 hold true, with the only exception of (A3) which we expect to hold when $n_{pret} \gg n_*$. We verify both (A3) and our theoretical claim (4) numerically. As shown in Fig. 1 below, (4) appears to hold (middle subplot) when (A3) is fulfilled (left subplot), in which case the credible intervals constructed from the MP also demonstrates valid coverage.

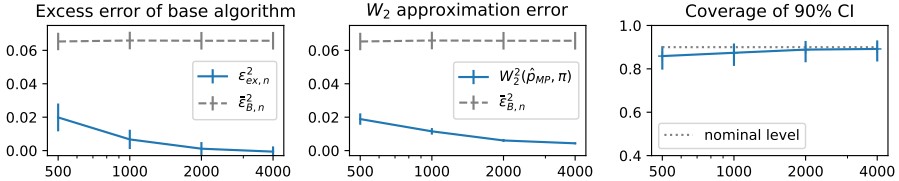

Figure 1: Result for the multi-task learning simulation, with $n_* = 20, m = 200$ and varying $n_{pret}$. Plotted are the mean and 95% confidence interval for each metric.

**Hyperparameter learning for GP regression.** We investigate if the proposed method may alleviate overfitting in empirical Bayesian hyperparameter learning. We evaluate our method on 9 UCI regression datasets adopted by Salimbeni and Deisenroth (2017), each subsampled to $n \in \{75, 300\}$ training observations. As shown in the full paper, our method

outperforms the empirical Bayes algorithm, bootstrap aggregation and a vanilla ensemble method based on initialisation randomness, in terms of test mean squared error, log predictive density and continuous ranked probability score (CRPS).

**Classification with tree and AutoML algorithms.** We apply our method to two base algorithms: *(i)* XGBoost (Chen and Guestrin, 2016) which implements gradient boosting decision trees (Friedman, 2001), and *(ii)* AutoGluon (Erickson et al., 2020), an AutoML system which performs stacking (Wolpert, 1992) using a variety of tree and DNN models. Both are highly competitive approaches that outperform deep learning methods (Grinsztajn et al., 2022; Shwartz-Ziv and Armon, 2022) but lack a natural Bayesian counterpart. We evaluate on 30 OpenML datasets adopted by Hollmann et al. (2022); for each algorithm we apply our method (IPB), and compare with bootstrap aggregation (BS) as well as the base algorithm without aggregation. The results are summarised in Table 1: for both base algorithms, our method improves over the base algorithm and its bagging variant.

Table 1: Classification experiment: average rank ($\downarrow$) of test metrics across 30 datasets.

| Metric | XGBoost | | | AutoGluon | | |
|---|---|---|---|---|---|---|
| | (Base) | + BS | + IPB | (Base) | + BS | + IPB |
| Log likelihood | 4.77 | 4.33 | 3.20 | 3.60 | 3.03 | **2.07** |
| Accuracy | 4.87 | 4.43 | 3.23 | 3.50 | 2.50 | **2.47** |

Table 2: Interventional density estimation: average rank ($\downarrow$) across all synthetic datasets.

| $n$ | PB | Ens. | NTKGP | BS | IPB |
|---|---|---|---|---|---|
| 100 | 3.6 | 1.9 | 5.0 | 3.1 | **1.0** |
| 1000 | 4.0 | 1.9 | 5.0 | 2.4 | **1.2** |

Table 3: Results on the fMRI datasets. We drop PB and NTKGP as they are uncompetitive in Table 2, but add the baselines and the flow-based method in Khemakhem et al. (2021) for reference. Boldface indicates the best result ($p < 0.05$ in a $Z$ test).

| Metric | Linear | ANM | Flow | D + Ens | D + BS | D + IPB |
|---|---|---|---|---|---|---|
| CRPS | $.738_{\pm.10}$ | $.551_{\pm.01}$ | $.546_{\pm.02}$ | $.520_{\pm.00}$ | $\mathbf{.518}_{\pm.00}$ | $\mathbf{.518}_{\pm.00}$ |
| Med. Abs. Err | $.658_{\pm.03}$ | $.655_{\pm.01}$ | $\mathbf{.605}_{\pm.02}$ | $.609_{\pm.01}$ | $.611_{\pm.01}$ | $\mathbf{.604}_{\pm.00}$ |

**Interventional density estimation with diffusion models.** We evaluate our method on the task of interventional density estimation (Khemakhem et al., 2021), which can be seen as conditional density estimation with distribution shifts induced by the intervention. We apply our method to the algorithm of Chao et al. (2023) which is based on diffusion models, and compare with the following ensemble methods: parametric (PB) and nonparametric (BS) bootstrap, deep ensemble (Ens), and the method of He et al. (2020, NTKGP). We evaluate on *(i)* 8 synthetic datasets in Chao et al. (2023), and *(ii)* a set of real-world fMRI datasets in Khemakhem et al. (2021). Following the original papers, we report an average maximum mean discrepancy (MMD) metric for *(i)*, and the median absolute error for *(ii)*. To evaluate the *quality of uncertainty estimates*, we further report the coverage of credible intervals for interventional mean functions for *(i)* (in Appendix D.4 of the full paper) and CRPS for *(ii)*. As shown in Table 2–3, the proposed method (IPB) attains the best overall performance.

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
