# OpenReview forum: "On Uncertainty Quantification for Near-Bayes Optimal Algorithms"
_approximateinference.org/AABI/2024/Symposium — AABI 2024_

### Official Review · Reviewer_v3Cx · 2024-04-20
**Review: On Uncertainty Quantification for Near-Bayes Optimal Algorithms**

**Rating:** 8
**Confidence:** 3

**Review:**

This paper is about Uncertainty Quantification of Near-Bayes Optimal Algorithms which are algorithms that satisfy eq. (1) in this paper. The paper is packed with detailed results, and it took me quite some time to get through it as there is a lot of important details deferred to the appendix paper which on its own has a massive supplement.

I appreciate the authors including all of these details of mathematical proofs and numerical results, however, I also think that they packed too much information into too little space for this conference paper. After getting through it, the content of the paper is in my opinion a clear accept, but I think the authors did not communicate key idea(s) intuitively; they are over-complicating their communication. Although attempting to come with examples of when their approach is meaningful, such as examples with "practitioner working on a new image classification dataset", it is still unclear for the reader when this approach is applicable. I will start to list my strengths followed by weaknesses, but I believe this paper is a clear accept.

Here are the strengths:
- The paper seems mathematically solid and grounded with important findings.
- Although their general method is very close to bootstrapping, their theoretical and empirical behaviour is indeed more desirable. Thus I think they bring enough originality to the table in order for this paper to be novel.
- The experimental results speak for themselves; as they beat SOTA in a variety of applications. I would have love to see some predictive distributions in low dimensional examples to see how their uncertainty estimates look compared to existing methods in the main paper, but I needed to scroll all the way to appendix D.

Here are the weaknesses as I see them:
- I don't think the authors does a good job explaining eq (1) thoroughly enough. What is \epsilon_n ? The authors mention epistemic uncertainty (not enough samples), is the why it depends on n? What is the difference between P_\theta_0 and p_\theta_0 ? The answers to these questions might be obvious to the authors but certainly not for readers not directly familiar with this notation. I recommend a big "Figure 1" which pedagogically illustrates the process of this idea and maybe how it differs from previous ideas.
- What does "promising reports from past literature" mean? It is too abstract and the connection down to how the practitioner should act might very well be is lost for the reader.
- "The practitioner may then commit to the choice of A that corresponds to the smallest ϵ_n." How so? Can't ϵ_n be large but the expectation over the loss function small and vice versa?
- How does one deem a dataset *similar* to the one a practitioner is working on? Only in rare occasions are two datasets similar enough for many of these assumptions to hold, unless the argument is that this approach only works for the "usual suspect"-datasets and tasks. In most cases, a practitioners dataset can be very different from any dataset out there, so what happens then?
- The authors assume that "the past and present tasks can be viewed as i.i.d. samples from the distribution π". But this might not hold in most cases since the choice of a task very much depends on what other tasks have been carried out. At least, the authors should clarify when this is applicable. In their own example, they say that "a practitioner working on a new image classification dataset" which must depend on the fact that other practitioners also have done this and how they did it, and thus choosing that task was not independent from previous tasks.
- Does this approach only work for i.i.d. samples? It seems to be a key assumption in eq 2.
- The paper does not (promise to) provide a code repo to how the numerical results were generated. This would be preferred.

---

### Official Review · Reviewer_jxku · 2024-04-21
**The paper is built on top of "Martingale posterior distributions" and it explores the use of martingale posterior to estimate the epistemic uncertainty of a near-Bayes optimal estimator.**

**Rating:** 4
**Confidence:** 2

**Review:**

Strengths:
- The authors show the conditions that lead to a bound on the 2-Wasserstein distance between the unknown Bayesian posterior and the Martingale posterior.
- A series of experiments conducted validating the efficacy of MP for uncertainty estimation.

Weakness:
- It is not clear to me what the MP-Inspired Uncertainty Quantification algorithm offers in addition to the one in "Martingale posterior distributions" by Fong, et al.

Clarity: The clarity of the paper could be improved. I believe too much information compressed into six pages, which makes the reading experience difficult.

Originality:
- The main theoretical result (the bound on the distance between Bayesian posterior and MP) seems to be original.
Although, I do not see much of difference between the proposed algorithm and Algorithm 3 of "Martingale posterior distributions". In any case, I think the authors should cite Fong, et al, given that the algorithm appears to be very similar to theirs.

Significance:
- The variance estimation algorithm's results appear promising. However, since the uncertainty estimation algorithm appears to resemble previous work (as noted above), I'm uncertain about the unique impact of this study. It would be beneficial for the authors to clarify the similarities between these 2 algorithms.

Note: My main concern with this work is the originality of the variance estimation algorithm. While the results look interesting, if the algorithm already exists (and has not been cited properly), I would be inclined towards rejection. I would adjust my score after the rebuttal.

Minor:
- In Eq(1), different notations used for p_theta_0.
- In Eq(2), are all \hat{Z} coming from P_\hat{\theta_j}? Then \hat{z}_{j+1} ~ P_\hat{\theta_j} -> \hat{z}_{i+1} ~ P_\hat{\theta_j} for i ...
- The classification experiments are conducted for XGBoost, I wander how MP uncertainty estimation perform for a deep neural network. Would overfitting create a challenge for the algorithm?

---

### Official Review · Reviewer_uMGP · 2024-04-23
**Interesting approach to small sample uncertainty quantification in a multi-task setting, similar to bootstrap**

**Rating:** 8
**Confidence:** 4

**Review:**

--The paper considers quantifying uncertainty with small-sample learning in a multi-task setup.  This assumes that some data is drawn from an underlying parameterized distribution, and each ‘task’ corresponds to a different parameter.  So the learning problem (the ‘tasks’) are statistically similar such that learning over different but related tasks is advantageous. The problem studied in the paper is well motivated and characterizing small sample estimation in this context is challenging.  The proposed algorithm seems to be a variant of bootstrap ideas.  Overall the paper is interesting, although the numerical experiments don’t necessarily give a good sense of what happens as the sample size grows, what is the ‘small’ regime for a given problem, or whether the proposed sampler has any advantage outside of the small sample regime.


--The paper relates learning architecture to (approximate) martingale posteriors (MP), which applies under some assumptions and approximations.  This leads to MP-inspired algorithms for uncertainty quantification.  In general it may be difficult to prove the assumptions hold in cases other than well defined or relatively simple, although the algorithmic approach might still be useful in practice.   The paper shows that the approach should be useful for general Gaussian Processes, and this seems to be an important and practical case.


--The general approach presented in section 4 is very similar to bootstrap with a parametric model and appears to be a variation of these ideas.  This is explored in the attached full paper to some extent.


--In general there seems to be a parameter size mismatch.  Footnote 2: the overparameterized case, such as using neural networks, seems to be of the most interest but difficult to apply in practice?  How is the semi-metric d(theta,theta’) chosen?    Apparently the Bregman divergence is used (Remark 2, section 4).  Linking this with the underlying MP theory is clearly an interesting challenge!


--For the assumptions leading to eqn (3), this requires the gradient wrt known theta?

---

### Official Review · Reviewer_Tq3W · 2024-04-24

**Rating:** 5
**Confidence:** 2

**Review:**

To begin with, I have two concerns about the form of this submission:

* The appendix appears to be the full version of the submission. In particular, the submitted 6 pages seem to be a subselection of the writing of the full version! In my opinion, it is perfectly fine to defer detail to the appendix, but the submitted 6 pages and the appendix should form a cohorent whole. That the authors append the PDF of the full version, without modifications, as the appendix demonstrates a sense of laziness, in my opinion.

* The submission is highly technical and all proofs are deferred to the appendix. This is not necessarily a problem. However, without spending a significant amount of time spitting through the proofs of the appendix, which I would be happy to do for full conference submissions, I am unable to make any statements about the correctness of the results in the submission.

All in all, I am of the opinion that the non-archival track of AABI is not the right venue for this work. A better place would be a full conference submission, preferably theoretical in nature, where the reviewers review the proofs in detail. Nevertheless, I appreciate that the authors might want some early feedback on their work, before possibly submitting to another place. For these reasons, I will give a marginal reject, leaving it up to the other reviews and AC to decide whether the AABI non-archival track is the right place for this submission.

The remainder of this review will focus on the content of this submission.

------

I think this submissions is very interesting! Attempting to develop new methods for uncertainty quantification is very much worthwhile, in my opinion. Doing this by assuming that the algorithm under consideration is near-optimal in terms of average-case performance is an interesting way to go about that.

If I understand the submission correctly, the proposed algorithm is Algorithm 1, which is motivated by the theoretical developments before. There is one aspect I do not quite understand:

The writing seems to suggest that Algorithm 1 can be applied to all online algorithms which produce point estimates and which satisfy certain technical conditions. However, Algorithm 1 also uses samples from $p_{\hat \theta_j}$, a density at the current point estimate of the parameter. Doesn't that mean that the algorithm to which Algorithm 1 applies also requires some form of the density of the underlying data? How would this work for a neural network applied to some arbitrary data, where there is no density for the underlying data? Or is the requirement that the algorithm should be able to sample new data?

I like that the authors give examples of scenarios where their assumptions are satisfied.

The form of the assumptions and theoretical results look reasonable. However, without carefully going through the appendix, I cannot at all comment on the validity of the theoretical setup and results.

The experiments appear to be carefully setup and demonstrate promising results.

All in all, if this submission were to be accepted, despite my concerns about its form, I think it would generate some interesting discussion at the poster session.

---

### Official Review · Reviewer_nGNd · 2024-04-24
**Excellent work**

**Rating:** 8
**Confidence:** 4

**Review:**

This paper discusses the optimality of Martingale Posterior in theory, designs an algorithm on Grad Des that can express uncertainty and be scalable, and demonstrates the feasibility of the algorithm in different specific ML problems. The whole work is novel and complete.

Some suggestions or ideas:
1. Can you explain more clearly the specific implementation of \widehat{z}_{n_{j}: n_{j}+\Delta n} \sim p_{\widehat{\theta}_{j}} in **Algorithm 1** in each experimental case (especially those containing DNN)?
2. As the practical guide of (D)NN, although the article has mentioned the paper dealing with small samples, can you introduce more classic experimental examples?
3. Can you add some intuitive figures? After all, fMRI data is used, maybe the uncertainty in this experiment can reveal some facts.

---

### Official Review · Reviewer_AqHM · 2024-04-26
**Somewhat understandable advert for main paper that describes analysis of martingale posteriors and proposes an algorithm to construct uncertainty estimates for sequential learning algorithms**

**Rating:** 7
**Confidence:** 3

**Review:**

The paper analyses martingale posteriors, a method to construct posterior distributions over parameters sequentially using a possibly randomized learning algorithm that returns parameter point estimates that have been refined using the previous estimate and new data. They show that for algorithms that are near optimal in that they achieve posterior error that better matches the optimal estimator error with increasing data, the difference between the martingale and Bayesian posteriors is composed of higher order terms. They then propose an algorithm inspired by the martingale posterior to generate uncertainty quantification from learning algorithms that only produce point estimates. Experiments show better performance over baseline methods on a range of datasets.

Pros: The paper was somewhat clear to understand but due to some typos (e.g. $\bar{\epsilon}^{2}_{B,l}$ doesn't seem to match the form given in the main paper) some time was needed to make sure I could get the gist of the paper. The analysis and provided algorithm seem like an original and significant use of martingale posteriors (which I have no expertise in). The algorithm is generic enough that it would be of use to practitioners who seek to get some uncertainty estimates from their iterative learning algorithms that return point estimates. Examples of algorithms given in section 3 that fit their framework is a great addition as it gives a sense as to how their analysis and algorithm may be used in practice.

Cons: The major weakness of the paper would be the experiments section. Although using their technique results in increased performance over baselines the results don't provide any intuition as to how their algorithm works in practice. For a conference like AABI adding an experiment section that elucidates the capabilities and limitations of their approach would result in readers better understanding how the approach can be used and the practical implications of the analysis in section 3. Relatedly, a lack of a more comprehensive characterization of the algorithms for which their analysis applies makes it unclear as to which learning algorithms their results apply to. Although A3 gives some intuition of what algorithms these would be it is not clear how this condition would be verified for some arbitrary algorithm that fits the framework.

Overall, the paper provides important analysis of martingale posteriors and a practical algorithm to produce uncertainties though deeper understanding of the analysis and approach isn't clear to understand from the AABI paper. I believe this work would be of some interest to the AABI community.

---

### Meta-Review · Area_Chair_pbEZ · 2024-05-12

**Recommendation:** Accept (Poster)
**Confidence:** 4

**Metareview:**

This paper proposes a way to do uncertainty quantification based on point estimation algorithms. The key analysis is the martingale posterior proposed by Fong et al., 2021: the authors prove that under some conditions, the martingale posterior is close (in 2-Wasserstein distance) to the exact Bayesian posterior. Then, based on this, the authors proposed a simple sequential algorithm for building an ensemble. Experiments indicate that their proposed method is effective against standard baselines.

Most of the reviewers agree that this paper is of high quality. However, one reviewer mentioned that the proposed method resembles Algorithm 3 in Fong et al., 2021. Another reviewer noted that this submission seems "low-effort"; the authors simply append the full paper with a different style in the appendix. Moreover, the abstract is also missing.

I recommend acceptance. Nevertheless, the authors should seriously address all reviewers' comments.

---

### Decision · Program_Chairs · 2024-05-27

Accept